

# Confirmation of the southern African distribution of the marine sponge *Hymeniacidon perlevis* (Montagu, 1814) in the context of its global dispersal

Toufiek Samaai[1,2,3,4], Thomas L. Turner[5], Jyothi Kara[3], Dawit Yemane[6], Benedicta Biligwe Ngwakum[7], Robyn P. Payne[2] and Sven Kerwath[6]

[1] Biology Department, University of Cape Town, Rondebosch, Cape Town, Western Cape, South Africa
[2] Department of Biodiversity and Conservation, University of the Western Cape, Bellville, Cape Town, Western Cape, South Africa
[3] Department of Research and Exhibitions, Iziko Museums of South Africa, Cape Town, Western Cape, South Africa
[4] Oceans & Coasts Research, Department of Fisheries, Forestry and Environment, Cape Town, Western Cape, South Africa
[5] Ecology, Evolution, and Marine Biology Department, University of California, Santa Barbara, CA, USA
[6] Fisheries Research and Development, Department of Fisheries, Forestry and Environment, Cape Town, Western Cape, South Africa
[7] Department of Pharmaceutical Sciences, Tshwane University of Technology, Pretoria, Gauteng, South Africa

Corresponding author
Toufiek Samaai,
toufiek.samaai@gmail.com,
tsamaai@dffe.gov.za

## ABSTRACT

**Background**. Intertidal rocky shore surveys along the South African coastline (∼3,000 km) have demonstrated the presence and abundance of the encrusting orange sponge *Hymeniacidon perlevis* (Montagu, 1814), a well-known globally distributed species. After analysing the southern African populations, we gained a better understanding of the genetic structure of this now-accepted global species. Apart from confirming the presence of a single population of *H. perlevis*, we also determined its distribution in the southern African intertidal rocky shore ecosystem, compared its genetic diversity to congeners, predict its global distribution via environmental niche modelling, and discussed possible underlying mechanisms controlling the species' global distribution.
**Methods**. We surveyed the South African coastline and sampled sponges at 53 rocky shore sites spanning over 3,000 km, from Grosse Bucht south of Lüderitz (Namibia) to Kosi Bay on the east coast of South Africa. DNA sequences of the nuclear rDNA internal transcribed spacer (ITS1) and the COI mitochondrial gene were obtained from 61 samples and compared them to a world-wide sample of other *H. perlevis* sequences. Using environmental predictor variables from the global dataset BIO-ORACLE, we predicted the probability of global occurrence of the species using an ensemble of eight distribution models.
**Results**. South African specimens were found to be 99–100% identical to other populations of *H. perlevis* (=*H. sinapium*) from other world-wide regions. The presence of a single population of *H. perlevis* in southern Africa is supported by genetic data, extending its distribution to a relatively wide geographical range spanning more than 4,000 km along the temperate southern African coast. The predicted global occurrence by ensemble model matched well with the observed distribution. Surface temperature mean and range were the most important predictor variables.

**Conclusion**. While *H. perlevis* appears to have been introduced in many parts of the world, its origins in Europe and southern Africa are unclear.

## INTRODUCTION

*Hymeniacidon perlevis* (Montagu, 1814) is a common sponge in a wide variety of coastal habitats, occuring up to 3 m above the low tide line down to shallow subtidal habitats (*Gastaldi et al., 2018*; *Regueiras et al., 2019*; *Turner, 2020*; *Harbo et al., 2021*; *de Voogd et al., 2021*). A recent compilation of genetic and morphological evidence confirmed that the species has been found in temperate waters of all major ocean basins (Fig. 1; Table S1; see also *Turner, 2020*), and it has 18 synonymized names from various locations around the world (*de Voogd et al., 2021*). This very prolific encrusting orange sponge was first described from Devon, southwest England as *Spongia perlevis* Montagu, 1814. The type locality has very similar environmental conditions to the west coast of South Africa (*Smit et al., 2013*). The original description is very limited, with no figures of the diagnostic characters (see Montague, 1814, pg. 86).

In southern Africa, *Hymeniacidon perlevis* were first detected at the turn of the 20th century. *Stephens (1915)* described two species, collected during the Scotia expedition in 1904, from Saldanha Bay, False Bay and Hout Bay as *Halichondria caruncula* (*Bowerbank, 1858*) and *Leucophloeus styliferus Stephens, 1915*. *Halichondria caruncula* (*Bowerbank, 1858*) is regarded as a junior synonym of *H. perlevis* (*Ackers et al., 2007*; *de Voogd et al., 2021*), and *Leucophloeus styliferus Stephens, 1915*, is now accepted as *Hymeniacidon stylifera* (*Stephens, 1915*) (see *de Voogd et al., 2021*). *Penrith & Kensley (1970a)*, *Penrith & Kensley (1970b)*, *Day (1969)* and *Day (1974)* reported the presence of *H. perlevis* in South Africa based on the published works by *Stephenson (1939)*, *Stephenson (1944a)*, *Stephenson (1944b)*, *Stephenson (1948)* and *Stephenson & Stephenson (1972)*. Additionally, *Branch et al. (1994)*, *Branch et al. (2002)*, *Branch et al. (2007)*, *Branch et al. (2010)*, *Branch et al. (2016)* and *Branch et al. (2017)* listed *H. perlevis* in their field guides and recorded its distribution as occurring from Port Nolloth on the west coast to East London on the east coast. *Hymeniacidon perlevis* was first described in detail from the west coast of South Africa by *Samaai & Gibbons (2005)*. During the 2001 Saldanha Bay port survey (A Awad, L Greyling, S Kirkman, L Botes, B Clark, K Prochazka, T Robinson, N Kruger, L Joyce, 2002, unpublished data: Port biological baseline surveys: draft report Port of Saldanha, South Africa), the species was not recorded; no Porifera were included in the geographical analyses of *Emanuel et al. (1992)*, *Awad, Griffiths & Turpie (2002)* and *Turpie, Beckley & Katua (2000)*. *Hymeniacidon perlevis* was reported by *Penrith & Kensley (1970a)* in the Lüderitz intertidal zone, and by *Kreiner et al. (2019a)*, *Kreiner et al. (2019b)* and *Kreiner et al. (2019c)* at Grosse Bacht, Diaz Point, Patrysberg, Mile 4, Badewane and Möwe Bay in Namibia. From Rocky Point to the Kunene River north of Möwe Bay, the species was not recorded (*Penrith & Kensley, 1970b*;

*Kensley & Penrith, 1980*; *Kreiner et al., 2019a*; *Kreiner et al., 2019b*; *Kreiner et al., 2019c*). In South Africa *H. perlevis* is not listed as an introduced species (*Robinson et al., 2005*; *Mead et al., 2011*; *Mead et al., 2013*; *Branch et al., 2017*), but the global distribution of the species has been attributed to maritime traffic (*Gastaldi et al., 2018*; *Schwindt et al., 2020*; *Turner, 2020*; *Harbo et al., 2021*).

Some sponges that were previously believed to have widespread distributions have been shown to be comprised of multiple cryptic species (*Xavier et al., 2010*; *de Paula et al., 2012*; *Pérez-Portela et al., 2013*). In addition, the larvae of *Hymeniacidon perlevis* are lecithotrophic with a relatively short planktonic life, resulting in low dispersal capacity (*Maldonado, 2006*; *Xue, Zhang & Zhang, 2009*). Biofouling of historical and modern vessel hulls and on shells of shellfish that were transferred between aquaculture facilities has been suggested as a likely mechanism for the transfer and introduction of this species (*Schwindt et al., 2020*; *Turner, 2020*; *Harbo et al., 2021*), but the origin and possible sequence of the introductions remain unclear. When dealing with potentially introduced species, reliable taxonomy is essential. The difficulty in detecting introduced sponge species has implications that go beyond systematic research, affecting ecological studies and management initiatives.

*Hymeniacidon perlevis* is recognized as a morphologically uniform species throughout its distribution (*Turner, 2020*). However, because the species is geographically widespread and there is a potential for considerable population structure due to the alleged limited dispersal capabilities, we were interested in how morphological uniformity aligns with genetic uniformity.

In the present study, we employed the analysis of two molecular markers, the mitochondrial cytochrome *c* oxidase subunit I (COI) and ribosomal ITS subunit to add to our understanding of the genetic structure and haplotype diversity of this species within southern Africa and globally and to evaluate the effects of geographic distance and connectivity in this conspicuous widespread species.

Environmental niche modelling was conducted to investigate the factors that dictate the distribution and the drivers of genetic structure of *H. perlevis* globally. While the origin and processes of spread are speculative at this stage, we discuss the most likely scenarios.

## MATERIALS & METHODS

### Museum material

*Stephens (1915)* material, *Hymeniacidon caruncula* (*Bowerbank, 1858*), NMSZ 1921.143.1443, from False Bay and Saldanha Bay, and *Leucophloeus stylifera Stephens, 1915*, Syntype, NMSZ 1921.143.1443, from Saldanha Bay, were acquired on loan for comparative studies. The specimens are kept in 70% ethanol in the Department of Natural Sciences, National Museums Collection Centre in Edinburgh, United Kingdom. The holotype of *Hymeniacidon sublittoralis Samaai & Gibbons, 2005*, SAM-4903, is preserved in 70% ethanol and deposited at the Iziko Museums of South Africa.

South African *Halichondria caruncula* recorded by *Stephens (1915)*, *Leucophloeus styliferus*, and *H. sublittoralis* were examined and compared with the South African and Namibian specimens of *H. perlevis*.

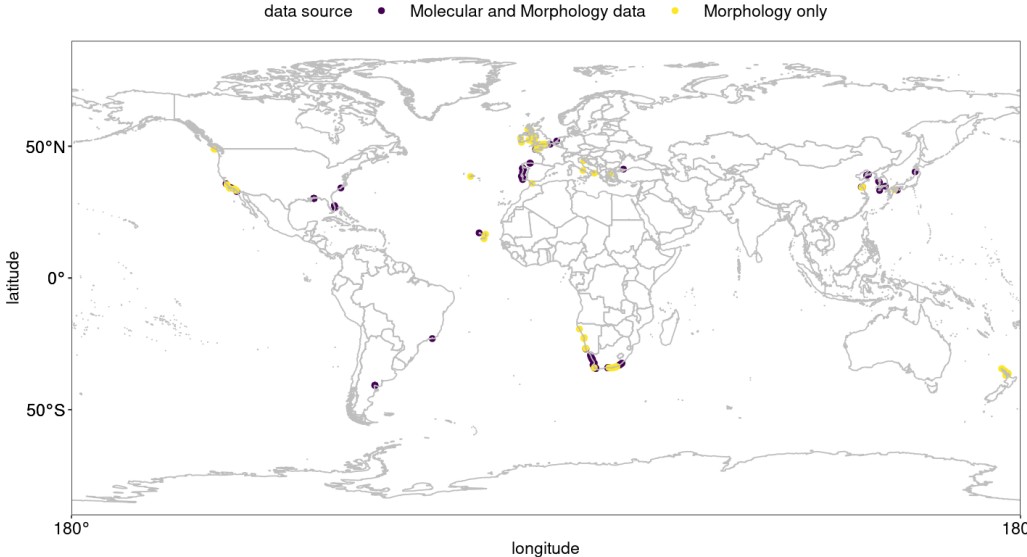

**Figure 1** **Global occurrence records of *Hymeniacidon perlevis* based on morphological and molecular data.** R, an open source program, was used to produce the map (*R Core Team, 2021*).

## Sample collection

Between 2015 and 2019, field surveys were conducted during low tide at 53 intertidal habitats along the South African coast from Port Nolloth (Benguela Current system, 26°44′7.05″S; 15°5′43.38″E) to Kosi Bay near the Mozambique border (Agulhas Current system, 26°55′46.21″S; 32°52′41.23″E) (Fig. 2; Table S2). Dr. Maya Pfaff collected a sponge sample from Grosse Bucht south of Lüderitz (Namibia) in 2019 as part of the Department of Forestry, Fishries and Environment (DFFE) (South Africa) and Ministry of Fisheries and Marine Resources (Namibia) joint rocky shore monitoring program (*Kreiner et al., 2019a*; *Kreiner et al., 2019b*; *Kreiner et al., 2019c*), and it was identified as *H. perlevis*. This sample was included in this study. We were not able to sample the rocky intertidal area between Bettys Bay and Knysna (Fig. 2). Sponges were collected from the intertidal rocky shores by removing a representative piece of the animal. Observations on appearance in life, habitat type and depth were recorded *in situ* (Fig. 3). Colour photographs were taken *in situ* (Fig. 3) and in the laboratory. Upon collection, specimens were stored in 96% ethanol and processed for histological examinations according to *Samaai & Gibbons (2005)*. Spicule dimensions are given as the mean length (range) × mean width (range) of 20 spicule measurements.

## Molecular analyses

DNA was extracted from 54 representative tissue samples across the three major biogeographic provinces using the E.Z.N.A Tissue DNA kit according to the manufacturer's protocol (Omega Bio-Tek). A fragment of the mitochondrial cytochrome *c* oxidase subunit I (COI) was amplified using primers LCO–1490 (5′—GGT CAA CAA ATC ATA AAG ATA

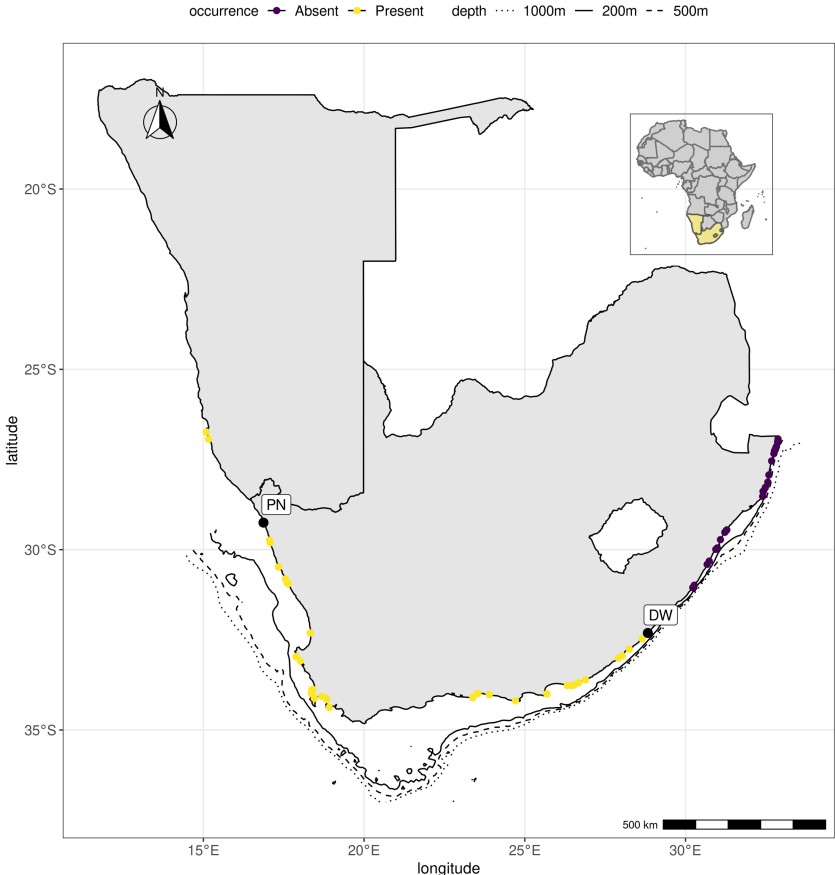

occurrence — Absent — Present    depth ···· 1000m — 200m - - 500m

**Figure 2** *Hymeniacidon perlevis* **range along the southern African coastline.** The presence of *H. perlevis* is shown by yellow dots, whereas the absence is indicated by black dots. Table S2 lists the locations that were surveyed. R, an open source program, was used to produce the map (*R Core Team, 2021*).

TTG G—3′) and HCO–2198 (5′–TAA ACT TCA GGG TGA CCA AAA AAT CA–3′) (*Folmer et al., 1994*).

Polymerase chain reactions (PCR) were performed in volumes of 25 µl containing 12.5 µl Taq, 0.5 µl of each primer (10 mM), 1 µl of BSA, 5 µl of DNA template and 5.5 µl H$_2$O. The cycling profile included an initial denaturation step (3 min at 94 °C), 40 cycles of denaturation (30 s at 94 °C), annealing (20 s at 45 °C) and extension (1 min at 72 °C), and a final extension step (10 min at 72 °C). The amplified DNA was purified with a PCR Clean-Up Kit according to the manufacturer's protocol. The final DNA product was sequenced in both directions on an Applied Biosystems 3730xl DNA Analyzer (see *Teske, Bader & Golla, 2015* for Standard protocols), and the obtained chromatogram was edited using MEGA11: Molecular Evolutionary Genetics Analysis version 11 (*Tamura, Stecher & Kumar, 2021*). All the sequences were deposited in GenBank (NCBI; *Benson et al., 2018*) under the accession numbers ON062377–ON062402 (see Table S3). No amplification product for *Halichondria caruncula* (*Bowerbank, 1858*) sensu *Stephens (1915)*, *Hymeniacidon stylifera*

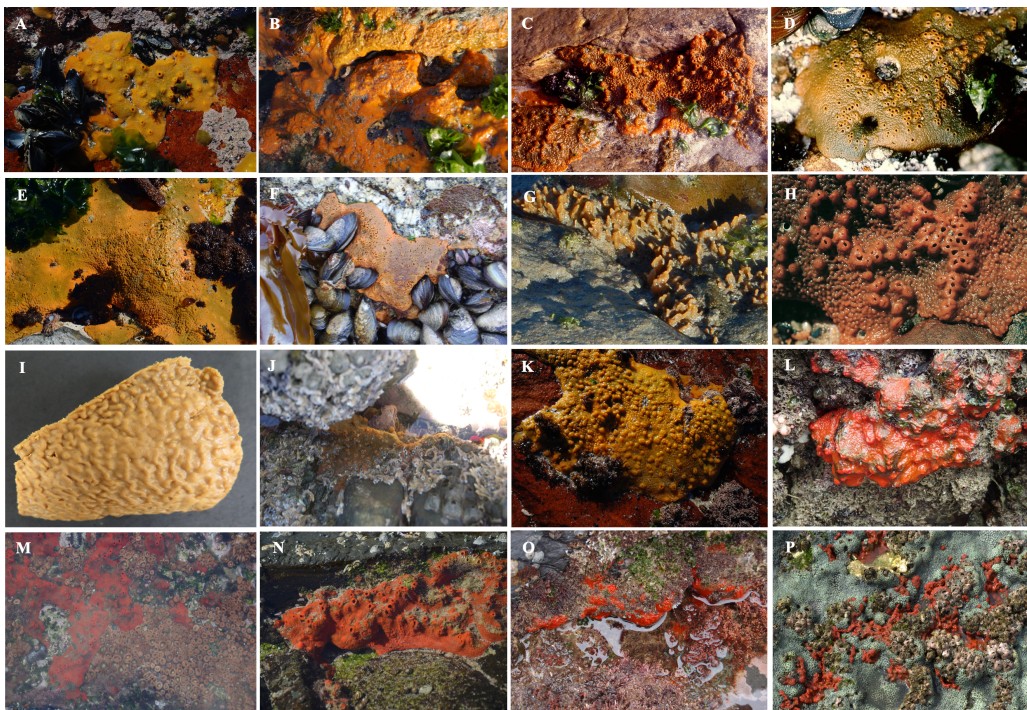

**Figure 3** Images of *Hymeniacidon perlevis* collected from different intertidal rocky shores along the South African coastline. (A) Groenrivier mund (photo credit: Prof. George Branch); (B) Moon Bay (photo credit: Dr Toufiek Samaai); (C) Elands Bay (photo credit: Prof. George Branch); (D) Jacobs Bay (photo credit: Dr Toufiek Samaai); (E) Springfontein (photo credit: Prof. George Branch); (F) Cape Peninsula Greenpoint (photo credit: Dr Toufiek Samaai); (G) Stand (photo credit: Dr Toufiek Samaai), (H) Kommetjie (photo credit: Prof. George Branch); (I) Bettys Bay; (J) Tsitsikamma (photo credit: Dr Toufiek Samaai); (K) Dwesa (photo credit: Prof. George Branch); *Tedania anhalens* from (L) Dwesa, (M) Coffee Bay (photo credit: Dr Toufiek Samaai), (N) Hluleka (photo credit: Prof. George Branch), (O) Port St Johns (photo credit: Dr Toufiek Samaai) and (P) Sodwana Bay (photo credit: Dr Toufiek Samaai). Appendix S7: Written permission from George Branch to publish Fig. 3 under the CC BY 4.0 license.

(*Stephens, 1915*) and *Hymeniacidon sublittoralis Samaai & Gibbons, 2005* from South Africa could be obtained.

## Alignment, phylogenetic analysis, and Haplotype networks

The raw sequence data of the forward and reverse sequences were trimmed by removing uncertain sites. The forward and reverse sequences were then aligned using ClustalW (*Thompson, Higgins & Gibson, 1994*) in MEGA 11 (*Tamura, Stecher & Kumar, 2021*). Sequences were blasted in GenBank (*Sayers et al., 2019*) and the maximum score and *E*-values (*Altschul et al., 1990*) were used to select closely related specimens. The COI data set was checked for the potential occurrence of nuclear pseudogenes using the genetic code for invertebrate mitochondria, to detect frame-shift mutations, which would indicate that these sequences originate from a non–functional gene region, were identified. Sequences were compared to published data of *Hymeniacidon* sponges (see Table S3 in *Turner, 2020*), and thus sequences were jointly analysed with the *Hymeniacidon* data set used by *Turner (2020)* in MEGA 11 (*Tamura, Stecher & Kumar, 2021*). This included public sequences

previously identified as *H. sinapium* and *H. heliophila*, which according to *Turner (2020)* are part of the single global species complex, *H. perlevis* (see Appendix S2–S6). The dataset compiled by *Turner (2020)*, together with the South African sequences, was then used in the phylogenetic analysis and haplotype network. To see if the South African samples formed a distinct monophyletic clade in comparison to congeneric samples from other locations, we constructed a phylogenetic tree in MEGA 11 using Maximum Likelihood (ML) with the Tamura-3 parameter (T92), which was selected by the inbuilt model generator. Evolutionary distances were computed employing the Tamura-3 parameter (*Tamura, 1992*), and support for individual nodes was based on 1000 nonparametric bootstrap estimates (*Felsenstein, 1985*). The T92 distances were also used to compare levels of genetic differentiation between the sequences generated in this study and the published *Hymeniacidon* sequences (see Table S3 in *Turner, 2020*).

DnaSP 5.091 was used to evaluate haplotype (h) and nucleotide ($\pi$) diversities for individuals collected at the same location (*Lourenço et al., 2017*). Populations were divided into six groups according to world-wide presence (*Turner, 2020*; *de Voogd et al., 2021*). These were East Asia, North America (Pacific), North America (Atlantic), South America (Atlantic), Europe and South Africa. The South African populations were divided into three groups according to the national bioregional classification (see *Sink et al., 2019*). These include the Namaqua, Southern Benguela and Agulhas ecoregions. To determine how genetic variation is divided between groups, among locations within groups and within locations, the six and four groups described above were designated a priori following *Lourenço et al. (2017)*.

A median-joining haplotype network was produced using the minimum spanning method (*Bandelt, Forster & Röhl, 1999*) as implemented in Popart (*Leigh & Bryant, 2015*). This analysis requires all included sequences to be the same length, so some sequences were trimmed whilst others were excluded. Alignments in the global dataset were 574 bp at CO1 ($n = 115$ sequences); the ITS alignment was 539 bp ($n = 512$). Alignments were longer when newly collected South African data were analyzed alone: 582 bp at CO1 ($n = 29$); 798 bp and ITS ($n = 11$). Sequence alignments were produced in Codon Code v.9 (CodonCode Corporation).

## Ecological niche modelling of *H. perlevis* distribution

To gain a better perspective on the realized distribution of *H. perlevis* globally, ensemble species distribution modelling was applied (Appendix S1).

*Occurrence data.* For this purpose, occurrence/encounter data from multiple sources were compiled, including records from the World Porifera Database (WPD), GenBank, literature and South African observations. The location data were checked by Toufiek Samaai and Thomas Turner and only valid occurrence records were included in this study (Table S1). Bias, dubious and unverified data were excluded.

Because the data are occurrence/encounter data, background absence (pseudo-absence) data are required to apply standard correlative distributional models. Pseudo-absences were generated at random within the studies gridded spatial domain, with the thin layer of coastal area being globally generated. Given the environmental layers resolution ($5 \times 5$

nautical mile grid cells), the first two adjacent grid cells were considered. The number of pseudo-absences generated were determined using the recommendations of *Barbet-Massin et al. (2012)*, implemented in the R package SSDM (*Schmitt et al., 2017*), which was used in this study for ensemble species distribution modeling. Ensemble modeling was performed both with and without spatial thinning. Given that the encounter data are not globally uniform, spatial thinning, which is already implemented in the SSDM package, was used to deal with spatial bias (to reduce spatial bias due to non-random sampling while keeping most of the information).

*Niche modelling.* Multiple correlative statistical models are widely used to model the distributions of many taxa. The majority of these widely used models are already included in the SSDM package. For the ensemble modeling of the distribution of *H. perlevis*, eight correlative statistical models were considered: generalized linear model (GLM), generalized additive model (GAM), support vector machine (SVM), classification tree analysis (CTA), generalized boosted model (GBM), random forest (RF), multivariate adaptive regression spline (MARS), and artificial neural network (ANN). Appendix S1 contains a brief discussion of each of the eight distribution models. These models were trained on a randomly selected portion of the data (70%) and their prediction performance was evaluated using the hold-out set. Each model was fitted and evaluated four times to account for sources of variability due to random selection of training and evaluation sets as well as random selection of pseudo-absences. When analyzing classification models, multiple measures of performance can be used. For the purposes of this study, Kappa and area under the curve (AUC) were used. AUC usually has a value in the range of 0.5 to 1.

Models with AUCs of 0.5 are generally considered random classifiers, while those with values between 0.7 and 0.8 are considered fair classifiers, and those with values close to 1 are considered excellent classifiers (*Kleinbaum & Klein, 2010*). The inclusion of individual distribution models into the ensemble distribution modeling was based on whether the model had an AUC value greater than 0.7. To generate the ensemble species distribution map, all models with AUCs greater than 0.7 were pooled by weighting their predicted probability of occurrence by their AUC. The uncertainty map was also computed primarily to identify regions of high agreement and low agreement among the models considered, which correspond to low and high uncertainty regions, respectively. Uncertainty was computed as cell by cell variance in the predicted probability of occurrence by the models included in the ensemble. The response curves (partial effects) of each of the variables considered was generated by predicting the probability of occurrence for the variable of interest while keeping the remaining variables at their mean. This was done for each of the eight models considered (Appendix S1).

The intertidal area was delimited by extracting the coastal cells covering a range from $-2$ to 1 m from the General Bathymetric Chart of the Oceans (GEBCO) gridded bathymetric data set with a spatial resolution of 30 arc-seconds (http://www.gebco.net/).

Variable importance was computed on the holdout set. The amount of correlation changes between predicted values before and after permuting (reshuffling) a variable was

used to measure its importance (expressed in percentage).

$$I_r = 1 - max\left(cor(P_f, P_v), 0\right)$$

where $I_r$ is index of importance of a variable, $Cor$ is correlation coefficient, $P_f$ is prediction from the full model, $P_v$ is prediction after permuting/reshuffling the variable $v$. Partial effect of each of the predictor were computed by predicting the response variable for the variable of interest while holding the other predictors at their mean.

*Environmental variables.* BIO-ORACLE was used to download global and readily available environmental layers. The environmental layers used in this study considered the minimum, maximum, mean, and range of surface temperature, surface salinity, and surface current velocity. Given the limitations of the environmental layers resolution (5 × 5 nm grid) and the fact that we are dealing with intertidal/coastal invertebrates, only grid cells within 10 km of the coastline were retained. The environmental variables were checked for multi-collinearity using the variance inflation factor (VIF) before the distribution of *H. perlevis* was modelled.

A VIF value of one indicates an absence of multi-collinearity, but larger values typically indicate the presence of a problem. Variables with VIF values > 5 are generally considered to be linearly related, and in the context of regression, their variance of the estimated parameters will be large, and its parameter will be poorly estimated (*Hay-Jahans, 2011*). Variables having VIF greater than 5 were thus excluded from this analysis. The final set of variables retained were: mean surface temperature, range surface temperature, mean surface salinity, range surface salinity, and mean surface current velocity. Appendix S1 shows the layers of environmental variables used in the distribution modelling.

The following model was fitted to model occurrence of the sponges.

Model                                    formula
*occurrence   sponge$_{occ}$ $\sim T_{mean} + T_{range} + S_{mean} + S_{range} + V_{mean}$*

where *sponge$_{occ}$* is the occurrence is of *Hymeniacidon perlevis*; $T_{mean}$ and $T_{range}$ are the mean and range of coastal surface temperature respectively; $S_{mean}$ and $S_{range}$ are the mean and range of coastal surface salinity respectively; $V_{mean}$ is the mean coastal surface current velocity.

All the analysis, visualization and report generation were done in R (*R Core Team, 2021*). Multiple R packages were utilized for data processing, visualization, analysis, and summary of results including (*Alathea, 2015*; *Allaire et al., 2021*; *Henry & Wickham, 2020*; *Robinson, Hayes & Couch, 2022*; *Wickham, Chang & Henry, 2018*; *Wickham et al., 2021*; *Xie, 2021*).

## Material and acquisition

All recently collected voucher samples are housed at the Iziko Museum, Cape Town, South Africa under museum numbers SAMC-A091444–SAMC-A091463; MB-A094583–MB-A094599; MB-A094600–MB-A094614 (Table S3). Toufiek Samaai was granted permission to collect specimens during his various field excursions by the Department of Forestry, Fisheries, and Environment under Research Permits RES2014/DEA–RES2019/DEA.

## RESULTS

### Distribution of *Hymeniacidon perlevis* along the temperate southern African intertidal region

*Hymeniacidon perlevis* was found at 23 of the 53 locations sampled (Fig. 2; Table S2). The confirmed species range spans from Grosse Bucht, south of Lüderitz, Namibia on the southern African Atlantic coast to the Dwesa/Cwebe Marine Protected Area in the Agulhas region of South Africa's Indian Ocean (east) coast. *Kreiner et al. (2019a)*, *Kreiner et al. (2019b)* and *Kreiner et al. (2019c)* reported the species from Grosse Bucht, Diaz Point, Patrysberg, Mile 4, Badewane, and Möwe Bay in Namibia, but no samples were available to confirm the identifications, with the exception of the specimen collected from Grosse Bucht. The species represents the most conspicuous and common sponge on the intertidal shores of the temperate bioregion around the tip of southern Africa. The species was found mainly associated with intertidal rocky shores at locations both associated with high anthropogenic impact such as Saldanha bay, Table bay and False bay, all areas with commercial harbours and international shipping traffic, and remote, natural locations including the Tsitsikamma National Park, South Africa's oldest Marine Protected Area.

We only found *H. perlevis* covered by sediment in one location (Strand), with surface projections extending beyond the sandy layer (Fig. 3G). *Hymeniacidon perlevis* was also found in high nutrient concentration areas throughout the Cape Peninsula (*De Villiers, 2017*; T Samaai, 2017, pers. obs.), in the kelp forest near Bettys Bay at a depth of 15 m, and on the west coast of South Africa by *Stephens (1915)* at a depth of 25 m. Although the species was not sampled between Bettys Bay and Knysna, it is present at various locations between these areas (Prof. G Branch, pers. comm., 2021).

### Genetic analysis

Partial COI sequences were obtained for 29 specimens of *H. perlevis* (Table S3) with an alignment length of 691 base pairs. This data set reduced to two haplotypes differing by a single base pair (Fig. S1). The haplotype network of *H. perlevis* from South Africa is shown in Fig. 4. One haplotype was present only in the Namaqua region, while the other was shared across regions.

BLAST-n results revealed these sequences to be 99–100% identical to sequences previously identified as *H. perlevis* and *H. sinapium* from other regions (for example Portugal and California; note that *H. sinapium* is now considered a junior synonym of *H. perlevis*). This was further corroborated by phylogenetic reconstruction, with sequences of *H. perlevis* from South Africa grouping with those from other regions and *H. sinapium* in the maximum likelihood tree (Fig. S2), with weak maximum likelihood support. The *H. perlevis* sequences further showed about 97% similarity to *H. flavia Sim & Lee, 2003* (EF217333.1, Korea), which is the most closely related outgroup species (Fig. S2) (*Park et al., 2007*). A haplotype network of this global sample is shown in Fig. 5. We employed a COI haplotype network to visualize the southern African specimens in the context of a global sample of publicly available data. One of the two haplotypes from southern Africa was among the most common haplotypes globally, also present in samples from Florida, Brazil, Portugal, Turkey, Korea, and California (Fig. 5).
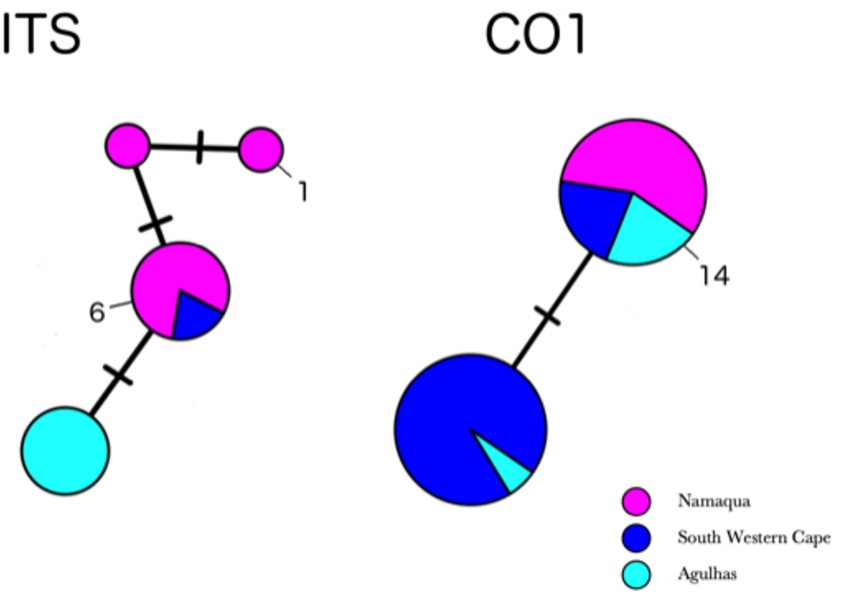

**Figure 4 Minimum-spanning genotype networks for two loci for South African samples.** Samples are coded by bioregion.

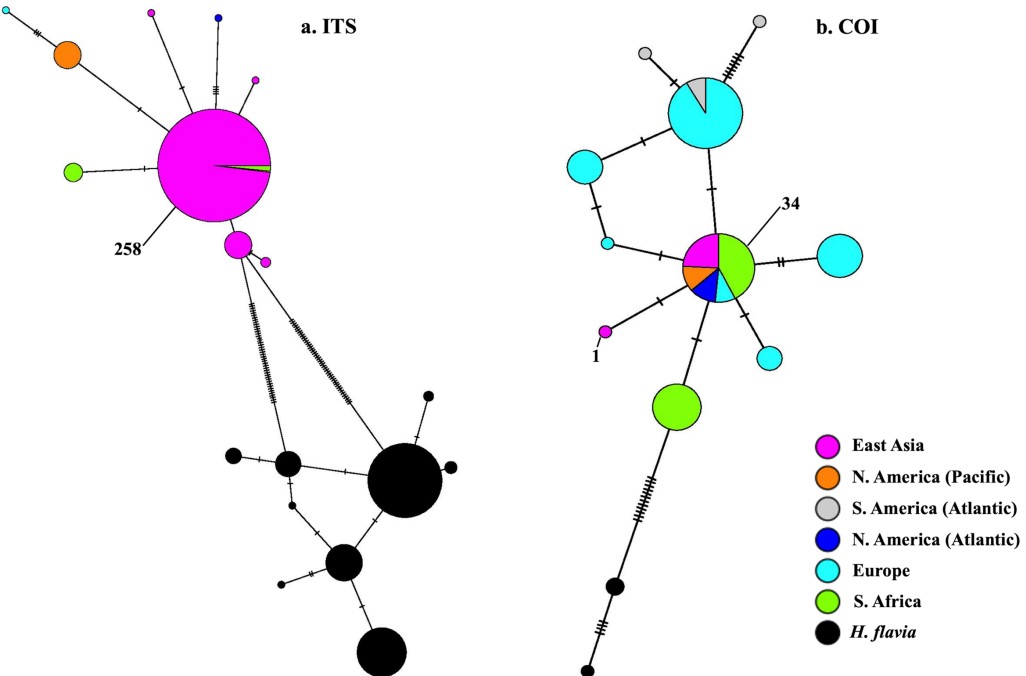

**Figure 5 Minimum-spanning genotype networks for two loci.** Samples are coded by collection location, regardless of whether they were identified as *H. perlevis, H. sinapium*, or *H. heliophila*. Closely related *H. flavia* are shown for comparison where available; all data for this species is from Japan and Korea.

In the global (539 bp) ITS alignment, four of the South African samples are identical to the most common haplotype, found in a sample from Northern California, five samples from Korea, and a large number of samples from Japan (Fig. 5). The other seven southern African samples differ from the most common haplotype by a single base pair (0.19% sequence divergence). Consistent with a previous analysis, the genetic variation within the sample of *H. flavia* is similar to the genetic variation in *H. perlevis*, despite the entire *H. flavia* sample being from Japan and Korea alone. When the South African ITS samples are analyzed alone, a longer alignment is possible (798 bp), and three variable sites are present (Fig. 4). One of these differentiates samples from Agulhas from the other regions.

## Environmental niche modelling

*Hymeniacidon perlevis* has a very wide geographic distribution ranging from the Northern Hemisphere to Argentina, as well as South Africa to the middle latitudes of New Zealand. Visual exploration of occurrence data for *H. perlevis* are shown in Fig. 1. As can be seen in Fig. 1, most of the occurrence records are from coastal area around the UK, New Zealand and South Africa, with the remaining records from different parts of the globe, such as harbours, mudflats and inlets (Table S1).

Figure 6 depicts the predicted distribution, as well as the probability of occurrence, of *H. perlevis* from each of the eight models and the ensemble (Appendix S1). The ensemble prediction on the raw and thinned data is shown in detail in Fig. 6. The projected likelihood of occurrence was not significantly different for models based on thinning or raw occurrence data when seen visually. The ensemble produced with the best models resulted in an accurate overall description of *H. perlevis* distribution, including its expanding front (Fig. 6).

Along southern Africa, the niche model predicted a distribution further north into Namibia and in South America from southern Chile into northern Peru (Fig. 6). In addition, the prediction indicated that suitable habitat could potentially be found along southern Australia and the south Island of New Zealand. While the probability of *H. perlevis* being present in the Mediterranean and British Columbia shores was high, towards Northeast America the predicted likelihood decreased. No suitable habitat was detected along the tropical West African coast, the Indo-Pacific region, the Arabian Peninsula or India. Mean surface temperature was the most important predictor of the distribution of *H. perlevis* globally (Appendix S1), followed by the range of surface temperature. Most of the models performed reasonably well with *AUC* mostly above 0.8.

## Taxonomy—species description

Systematic information with detailed morphological and spicule descriptions are provided below. The classification follows *Morrow & Cardenas (2015)*.

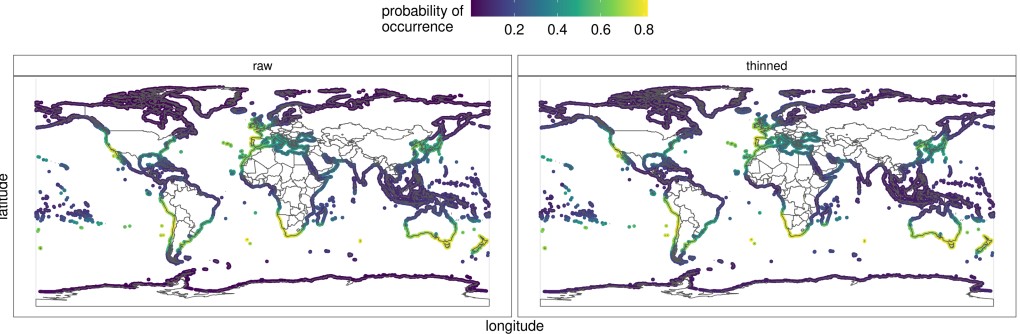

**Figure 6** Predicted global distribution for the orange/red encrusting sponge *Hymeniacidon perlevis* derived by averaging an ensemble of presence-absence algorithms. The thinned and raw data are shown. See Appendix S1 for report, data and model outputs. R, an open source program, was used to produce the map (*R Core Team, 2021*).

Phylum Porifera Grant
Class Demospongiae Sollas
Subclass Heteroscleromorpha Cárdenas, Perez & Boury–Esnault

Order Suberitida Chombard & Boury–Esnault
Family Halichondriidae Gray
Genus **Hymeniacidon** *Bowerbank, 1858*

**Type species**. *Hymeniacidon caruncula* Bowerbank, 1859: 286 (by subsequent designation; *Bowerbank, 1864*: 191) (this is considered a junior synonym of *Hymeniacidon perlevis* (*Montagu, 1818*: 86)).

*Hymeniacidon perlevis* (*Montagu, 1818*)
(Fig. 7AC; Table 1; Tables S4 and S5)

**Material examined**. Table S4

**Other material examined**. *Hymeniacidon perlevis* voucher specimens. SAM-H4904 (Ts 305), Jacobs Bay, near Saldanha Bay (32 31′S, 17 30′E), depth 3–5 m, collected by T Samaai, 20 October 1997. Ts 329, Ts 331, Ts 337, Ts 338, Ts 343c, Elands Bay (32 20′S, 18 20′E), depth 3–6 m, collected by T Samaai, 15 November 1997. Ts 359, Ts 370, Ts 381, Ts 391, Groenrivier (30 29′S, 17 20′E), depth 3 m. Collected by T Samaai, 20 December 1997.

*Hymeniacidon caruncula* (*Bowerbank, 1858*) sensu *Stephens (1915)*. NMSZ 1921.143.1443. Two fragments removed for loan. Station 479, False Bay shore 6th May 1904; Station 482, Saldanha Bay shore, 19 May 1904.

*Leucophloeus stylifera Stephens, 1915*. Syntype NMSZ 1921.143.1443. Two fragment removed for loan. Station 482, Saldanha Bay shore, 19th May 1904; Station 483, Entrance to Saldanha Bay, 45 m, 21 May 1904.

*Hymeniacidon sublittoralis Samaai & Gibbons, 2005*. Holotype. SAM-4903 (Ts 212), Vulcan Rock (34 04′S, 18 18′E), depth 27 m, collected by P Coetzee, 24 April 1996.

**Description**. A thin or thickly encrusting (Fig. 7A) to cushion-like sponge that varies greatly in form (Fig. 2). Diameter ranging from 5 cm long × 3 cm long × 4 cm thick to 14. five cm long × 8 cm wide × 6 cm thick; with processes of 1–4 mm high, 1–1.5 mm wide. In regions where there is considerably more wave exposure, this species is encrusting with a smoother surface. In sheltered or somewhat exposed locations, *H. perlevis* has upright processes that emerge from a basal mat. Surface variable, may be smooth and tuberculate, thrown into irregular folds, or covered with digitate processes. Oscules 0.5–1.5 mm in diameter, dispersed, level with the surface, or raised on low digitate processes. Firm, soft, fleshy texture that is compact and compressible. Live specimens of *H. perlevis* have distinct colour patterns being different shades of orange depending on geographical location (Fig. 2). Though this species can be blood red in other regions, in the current study, intertidal encrusting sponges of that colour were species of the genus *Tedania* (Fig. 2) (see also *Samaai & Gibbons, 2005*; *Ngwakum et al., 2021*).

**Skeleton**. The choanosomal skeleton, especially in the deeper regions, composed of a confused, disordered mass of styles, not organized into tracts (Fig. 7C). Towards the surface, tracts become ill-defined and with ascending fibres, ∼200 µm wide, with no separation between the primary and secondary tracts. The ascending tracts do not branch at the surface to form spicule brushes and tend to be vertically arranged. Numerous loose interstitial spicules. Large canals are present. The ectosomal skeleton consists of a dense tangential layer of spicules, ∼200–500 µm thick (Fig. 7C). Spongin scarce.

**Spicules**. Megascleres (Fig. 7B; Table 1; Fig. 8): styles smooth, straight, or slightly curved, thickest centrally, 250 (155–337) × 7 (7) µm, $n = 20$. Microscleres: absent.

**Habitat and distribution**. Found on the rocky intertidal areas, shallow subtidal reefs, mudflats on hard objects, harbours and inlets. Depth range 0–25 m (Table S1).

**Status**. Species may be native to southern Africa.

**DNA barcodes**. 691bp fragment of the universal mitochondrial cytochrome *c* oxidase subunit 1 gene, primer pair: LCO1490 and HCO2198 (*Folmer et al., 1994*). GenBank accession numbers ON062377–ON062402 & MT491492–MT491502 (see Table S3). 539 bp fragment of the ITS gene, Genbank accession numbers MT501787–MT501797 (see Table S3).

**Remarks.** The morphological features of *H. perlevis* and *H. caruncula* sensu Stevens (1915) were compared. *Hymeniacidon caruncula* sensu Stevens (1915) is similar to *H. perlevis*. In terms of spicule form, spicule size, external morphology, and colouring (Table S5), *H. sublittoralis* and *H. stylifera* differ from *H. perlevis*. *Hymeniacidon sublittoralis* is a thick, massive, erect, amorphous sponge, with numerous papillate processes that vary greatly in length. Surface smooth with various ridge-like structures, finely hispid and colour *in situ* yellow. Styles are large and thick with heads slightly subtylote, 394 (255–601) × 14 (14) µm (*Samaai & Gibbons, 2005*).

*Hymeniacidon stylifera* is easily distinguished from *H. perlevis* on spiculation; *H. stylifera* differs from *H. perlevis* by the larger size of the style megascleres, having a smooth surface and and have a very firm texture (Table S5).

The spicule size range of South African *H. perlevis* overlaps with specimens from Wales, Korea and Ireland (Fig. 8). A large spicule size range is found for the South African west

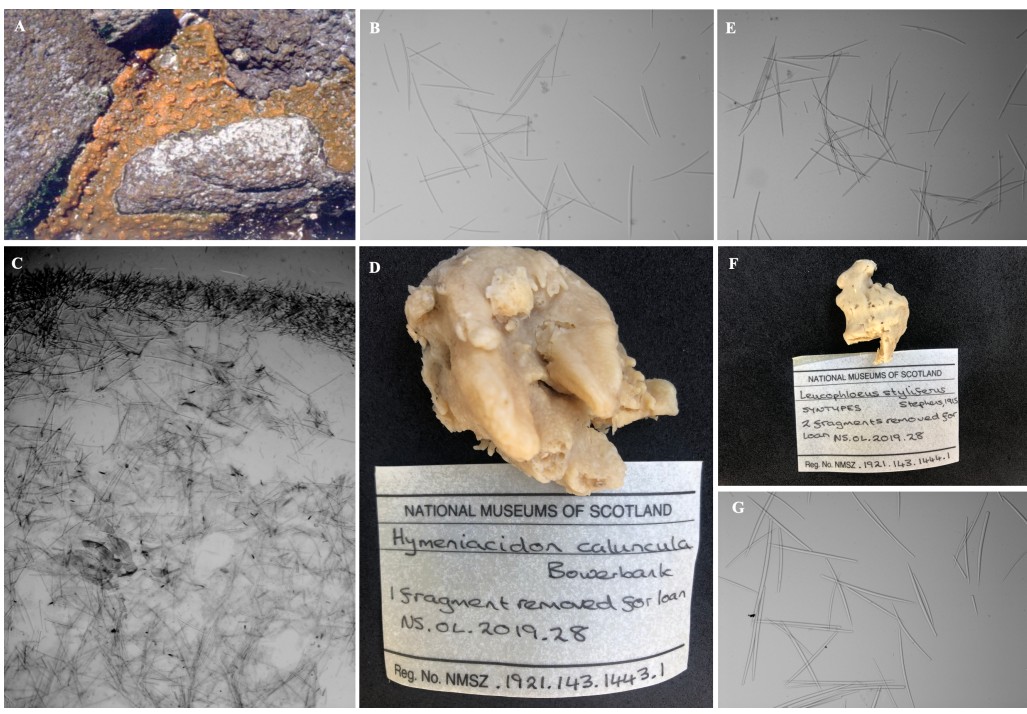

**Figure 7** *Hymeniacidon perlevis*, (A) *in situ*; (B) photomicrograph of spicule compliment, styles; (C) transverse histological section loose wispy tracts and paratangential layer in the ectosome; *Hymeniacidon caruncula* sensu *Stephens (1915)*, (D) *in situ*; (E) photomicrograph of spicule compliment, styles; *Hymeniacidon styliferus Stephens, 1915*, syntype, (F) *in situ*; (G) photomicrograph of spicule compliment, styles.

**Table 1** **Comparative micrometric data of spicules for voucher specimens of *H. perlevis* from South Africa.** Micrometric values in µm.

| Species | Specimen | Location | Large style | Medium style | Small style |
|---|---|---|---|---|---|
| *Hymeniacidon perlevis* | TS305 | Jacobs Bay | 337 × 7 µm | | 155 × 7 µm |
| | TS1163 | Knysna | 381 (336–437) × 5 µm | 270 (224–324) × 2.4 µm | 168 (146–190) × 2.4 µm |
| | TS1167 | Tsitsikamma | 386 (347–420) × 5 µm | 274 (246–308) × 2.4 µm | 167 (145–207) × 2.4 µm |
| | TS1189 | Robberg | 359 (336–370) × 5 µm | 285 (235–308) × 4.8 µm | 157 (140–196) × 4.8 µm |
| | TS2736 | Greenpoint | 375 (336–420) × 11.2 µm | 290 (246–314) × 11.2 µm | 162 (140–224) × 5.6 µm |
| | TS2736 | Dalebrook | 374 (336–420) × 11.2 µm | 290 (252–308) × 11.2 µm | 163 (140–196) × 5.6 µm |
| | TS2743 | Strand | 390 (358–427) × 11.2 µm | 294 (280–308) × 11.2 µm | 187 (168–190) × 5.6 µm |
| | TS2765 | Mazeppa | | 307 (280–336) × 11.2 µm | 144 (112–168) × 5.6 µm |
| | TS2935 | Groenrivier Mund | 416 (364–476) × 11.2 µm | 274 (224–336) × 5.6 µm | 194 (179–213) × 5.6 µm |
| | TS2963 | Rooiklippies | 493 (364–431) × 11.2 µm | 288 (246–336) × 11.2 µm | 198 (190–213) × 5.6 µm |
| | TS3359 | Dwesa | 378 (358–420) × 5.6 µm | 321 (302–336) × 5.6 µm | 187 (157–213) × 5.6 µm |
| | TS4860 | Haga Haga | 375 (336–403) × 5.6 µm | 321 (302–336) × 5.6 µm | 187 (157–213) × 5.6 µm |

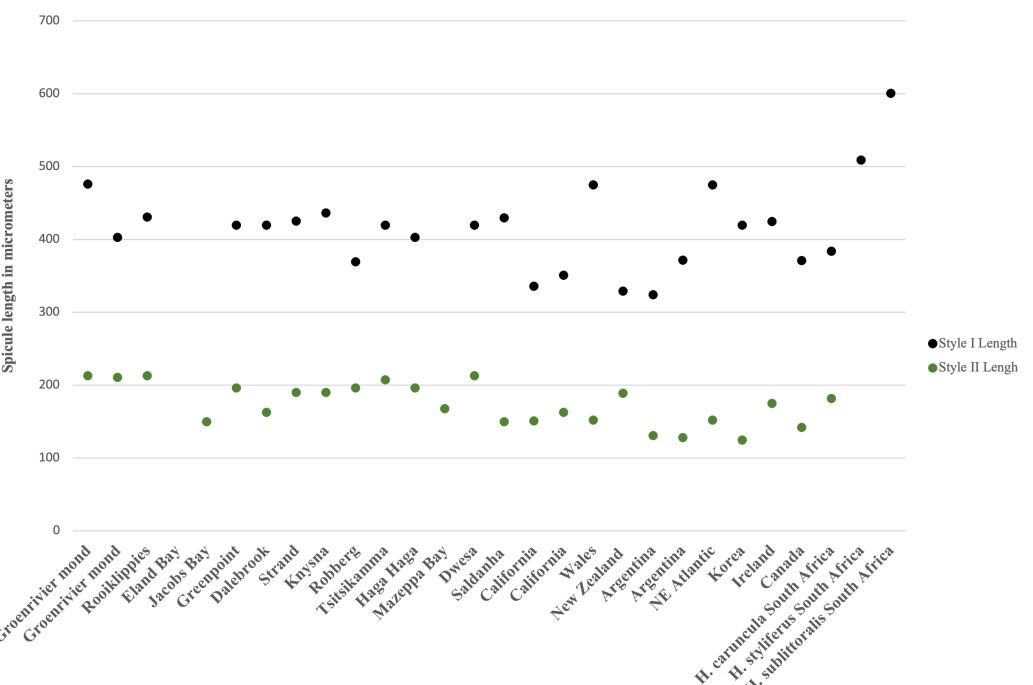

**Figure 8** **Megasclere spicule lengths for the South African and global species of *Hymeniacidon perlevis*.**
Each point represents the max and min spicule length for a specimen.

coast and Wales specimens (see Fig. 8). California and Canada specimens have a smaller spicule range, with lengths very similar to New Zealand and Argentina (see Fig. 8). No spicule lengths are available for specimens from Portugal, Spain, France, Brazil, Florida, China and the Black Sea. The South African specimens conform to all previous descriptions in terms of morphology and spicule complements, as well as in habit (*Bergquist, 1970*; *Ackers et al., 2007*; *Erpenbeck & Van Soest, 2002*; *Harbo et al., 2021*).

## DISCUSSION

Our genetic analysis confirms that *H. perlevis* is the most common and widespread sponge species on the temperate bioregion of southern Africa's intertidal rocky shores. Our results add a large extension to the range of this globally distributed species, confirming it to be the most widely distributed intertidal sponge known (*Turner, 2020*). Along the southern African coastline, the species is distributed across two biogeographic provinces, the cool-temperate west coast, and the warm-temperate south coast (*Emanuel et al., 1992*; *Samaai, 2006*; *Sink et al., 2019*). South Africa is an ideal region to study the effects of dispersal barriers and environmental gradients on species distribution (*Emanuel et al., 1992*; *Turpie, Beckley & Katua, 2000*; *Awad, Griffiths & Turpie, 2002*; *Bolton et al., 2004*; *Samaai, 2006*) and genetic patterns (*Teske et al., 2011*; *Teske, Bader & Golla, 2015*; *Zeeman et al., 2020*). The west coast of South Africa is permanently affected by the cold waters of the Benguela upwelling system, whereas the south flowing Agulhas current transports warm water along the east and south coasts of South Africa (*Lutjeharms, Cooper & Roberts, 2000*;

*Lutjeharms, 2006*; *Smit et al., 2013*). The region between Cape Point and Cape Agulhas represents a geographical break for several cool temperate and warm-temperate biota (*Emanuel et al., 1992*; A Awad, L Greyling, S Kirkman, L Botes, B Clark, K Prochazka, T Robinson, N Kruger, L Joyce, 2002, unpublished data: Port biological baseline surveys: draft report Port of Saldanha, South Africa; *Turpie, Beckley & Katua, 2000*) and is a driver of genetic differentiation (*Teske et al., 2011*). We found no evidence of this break reflected in the distribution of *H. perlevis* around southern Africa. The species occurs along the open coast and in embayments along both the Atlantic and the Indian Ocean coasts as a single population that lacks spatial genetic structure and exhibits little genetic variation.

*Duran, Pascual & Turon (2004)* reported a low level of genetic variation for partial COI sequences ($p = 0.0006$) among *Crambe crambe* sponges separated by distances from 20 to 3000 km, spanning from the western Mediterranean to the Atlantic coast. Despite the wide geographical coverage throughout the Indo-Pacific, *Wörheide (2006)* also found low nucleotide diversity among the sponge *Astrosclera willeyana* sensu lato ($p = 0.00049$). *Alex et al. (2012)* and *Alex (2013)*, on the other hand, found a much higher genetic diversity for *H. perlevis* across a very limited geographic region (500 km) along the Portuguese coastline (COI, $p = 0.00241$), suggesting considerable variability in this intertidal sponge species. Our research found that *H. perlevis* from southern Africa had low genetic diversity (COI: 0.00017) over a large geographic region (2,500 km), like *Crambe crambe* and *Astrosclera willeyana* sensu lato. We identified only two and four haplotypes for the COI and ITS genes, respectively, surveyed across the southern African distribution. The COI data presented here supports the hypothesis that just one *Hymeniacidon* species is found in southern Africa. Considering the geographic distance between sampled populations of more than 2,500 km, the observed mtDNA COI sequence variation is among the lowest for a diploblastic taxon to date as is the case for *Crambe crambe* and *Astrosclera willeyana* sensu lato, adding to the mounting evidence of general mtDNA conservation in sponges (*Duran, Pascual & Turon, 2004*; *Wörheide, 2006*).

The eastern limit of the southern African distribution of *H. perlevis,* the Dweza/Cebe MPA, falls into the transition zone between the temperate and sub-tropical biogeographic provinces. The continental shelf in this region gradually widens from north to south, deflecting the warm Agulhas Current away from the coast, limiting its influence on coastal biota. The northernmost breaks in this region have been identified in the Central Wild Coast (Transkei region in the region of Mbashe) and the southernmost breaks were reported near Algoa Bay (*Teske et al., 2011*). The subtropical and tropical Indian Ocean waters to the northeast of this area may be outside the environmental envelope of *H. perlevis*. In this bioregion the species is substituted by *Tedania* sp., the most conspicuous and common sponge in the subtropical rocky intertidal.

*H. perlevis* has a large ecological niche and can survive in intertidal and subtidal habitats with different substrata (*Turner, 2020*; *Harbo et al., 2021*) and is able to withstand large fluctuations in environmental conditions. As there is no obvious shift in habitat at the eastern limit of its distribution, sea temperature may limit its distribution. This is corroborated by the niche modelling, as it identifies sea temperature mean and range as the best predictor variables for the global distribution of this species. The model results

broadly reflect the distribution of the species along the temperate coastlines of all major ocean basins and accurately represent the eastern limit of the South African distribution, but the actual confirmed distribution of this species cannot be explained by sea surface temperature (SST) limitations alone.

There is little genetic variation in the molecular markers employed, so the structure of the haplotype network provides little indication on the historical biogeography of the species. We concur with several previous authors (*Turner, 2020*; *Harbo et al., 2021*) that *H. perlevis'* distribution is best explained by anthropogenic activity. Because larvae have a very limited chance of long-distance survival due to their short free-swimming phase, the vectors for this species are likely the transfer of mature adult colonies on ship hulls, shells and other objects in aquaculture activities (*Turner, 2020*; *Harbo et al., 2021*), or larvae transported in ballast water (*Duran, Pascual & Turon, 2004*). Due to this species' brief larval period, hull fouling, adult fragmentation and resettlement appear to be the most plausible mechanisms for long-distance invasions (*Turner, 2020*). Ship traffic between the two largest distributional nodes of this species, Europe and South Africa, has been going on for more than 600 years, much longer than between the other far-flung distributional nodes, and the species' wide distribution in these locations makes them the most likely places of origin.

The species was first described in Europe in the early 1800s and in South Africa in the early 1900s, more than 300 years after regular shipping commenced between the two areas. In southern Africa, the species occurs in unpopulated remote areas away from major ports on the open coast and in marine protected areas, but multiple introductions with shipwrecks as well as gradual range extension, as documented in other areas (*Turner, 2020*) might explain this distributional pattern. Further exploration with more rapidly evolving markers, such as microsatellites and SNPs, might aid to elucidate the history of the distribution of *H. perlevis*.

## CONCLUSIONS

Our work builds on a number of previous studies of *H. perlevis* and confirms that the species is present and widespread in southern Africa. Environmental niche modelling as well as the eastern range limit in South Africa, which coincides with the biogeographic break between temperate and subtropical waters, suggest that sea surface temperature is likely the most important limiting factor for this highly adaptable global species. *Hymeniacidon perlevis* is most likely an exotic species in many parts of its current distribution, introduced by shipping and other human-mediated activities. The origin of the species remain unclear, but it most likely originates from Europe or South Africa, where it is widely distributed across various habitats. Further molecular studies, increased systematic sampling, and monitoring, is required to clarify the origin of the species, mechanisms of its spread and its potential for negative impacts in areas of introduction.

## ACKNOWLEDGEMENTS

We thank Dr Maya Pfaff, Mrs Liesl Janson, Dr Tanya Haupt, Mr Laurenne Snyders, Ms Nicolette Naidoo, Mr. Imtiyaaz Malick, for their assistance during field work, and Iziko museum for the use of their Genetics laboratory. Dr Maya Pfaff and Dr Stephanie de Villiers are thanks for allowing me to participate in their field surveys around South Africa. The intertidal survey reports and species distributional data were provided by Dr. Anja Kreiner of Namibia's Ministry of Fisheries and Marine Resources. We thank Dr Wayne Florence, Director Research and Exhibitions and Mr. Dylan Clark curator of Marine invertebrate and fish collections at Iziko and all members of the Marine Biology Unit for their support. The senior author would also like to thank Mrs Liesl Janson for the histological preparations. Mrs. Ngwakum's MSc was supervised by Prof. Peter Teske, who is thanked for assisting and guiding Toufiek Samaai with his genetics research. We thank Prof. George Branch for the images of *H. perlevis* used in Fig. 3 and for his fruitfull discussions. We thank the three reviewers for their constructive comments that helped us improve the manuscript.

### Funding

This work was supported by the South African National Department of Forestry, Fisheries and the Environment, Oceans and Coasts Research, the National Research Foundation (grant numbers 129932) and forms part of the Marine Biodiversity Program. The funders had no role in study design, data collection and analysis, decision to publish, or preparation of the manuscript.

### Grant Disclosures

The following grant information was disclosed by the authors:
South African National Department of Forestry, Fisheries and the Environment, Oceans and Coasts Research, the National Research Foundation: 129932.
Marine Biodiversity Program.

### Competing Interests

The authors declare there are no competing interests.

### Author Contributions

- Toufiek Samaai conceived and designed the experiments, performed the experiments, analyzed the data, prepared figures and/or tables, authored or reviewed drafts of the article, lab Work—DNA extractions, PCR; Sampled the intertidal region; Species Identifications, and approved the final draft.
- Thomas L. Turner performed the experiments, analyzed the data, prepared figures and/or tables, authored or reviewed drafts of the article, provided data used in the manuscript; Generates Haplotype networks, and approved the final draft.
- Jyothi Kara conceived and designed the experiments, analyzed the data, authored or reviewed drafts of the article, lab work—DNA extractions, PCR, and approved the final draft.

- Dawit Yemane analyzed the data, prepared figures and/or tables, authored or reviewed drafts of the article, ran the Ecological and Environmental prediction models, and approved the final draft.
- Benedicta Biligwe Ngwakum conceived and designed the experiments, performed the experiments, authored or reviewed drafts of the article, lab work—DNA extractions, PCR, and approved the final draft.
- Robyn P. Payne conceived and designed the experiments, performed the experiments, authored or reviewed drafts of the article, lab Work—DNA extractions, PCR; Sampled the intertidal region, and approved the final draft.
- Sven Kerwath performed the experiments, analyzed the data, authored or reviewed drafts of the article, sampled for the sponges on the various field trips, and approved the final draft.

### Field Study Permissions

The following information was supplied relating to field study approvals (*i.e.*, approving body and any reference numbers):

Field work was approved by the Research Directorates of the Department of Fisheries, Forestry and Environment, Oceans and Coasts Research (research permits: res2014/dea–res2019/dea).

### DNA Deposition

The following information was supplied regarding the deposition of DNA sequences:

The South African COI sequences are available at GenBank: ON062377–ON062402 (Table S2).

The sequences of *Turner (2020)* are available at GenBank:

- COI: EF217333–EF217335; EF519629–EF519632; HM035983–HM035985; JX477015–JX477045; KF225481–KF225482; KY492551–KY492584; MG885802–MG885805; EF217329–EF217332; HM035986; HQ829181; KF192342; KP136744; MT007958; MT007960; MT007959; MT007958, MT001298, MT006362; MT007959, MT007960.

- ITS: EF217362–EF217364; AB373172–AB373186; EF217361; EF217360–AB373171; JQ658455–JQ658473; KT880468; JF824794–JF824787; AB373170–AB373171.

### Data Availability

The raw data is available in the Supplemental Files.

### Supplemental Information

Supplemental information for this article can be found online at http://dx.doi.org/10.7717/peerj.14388#supplemental-information.

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
