# Peer review of "Confirmation of the southern African distribution of the marine sponge Hymeniacidon perlevis (Montagu, 1814) in the context of its global dispersal"

_PeerJ, doi:10.7717/peerj.14388_

## Round 0.1 · original submission · Major Revisions

Two of the three reviewers were positive about your manuscript and suggested only relatively minor revision. The third reviewer, however, had a number of serious criticisms. In my view, a major revision of your manuscript can address most of the concerns that reviewer 3 raised, so I am inviting you to revise and resubmit. You may wish to follow reviewer 3's suggestion that your MS focus on the confirmation of Hymeniacidon perlevis's presence in southern Africa if their concerns about the rest of the paper cannot be addressed convincingly.

Incidentally, I noticed that blue & red are the wrong way round in the legend of Fig. 2, and depth needs units.

·

Basic reporting

All ok; a few typos and acronyms not spelled out the first time (see edited report)

Experimental design

All meet accepted professional standards

Validity of the findings

All ok

Additional comments

1. Some comment on the broad (=approximate) nature of the modeled physical parameters (temperature, salinity, current velocity, particularly means which need to be referenced since climate data are based on means) may be warranted. Extremes are much more important, especially to intertidal/shallow subtidal organisms, e.g., witness summer 2021 where significant numbers of intertidal organisms cooked in the heat. Ref. lines 320 and 613.
2. A huge amount of data were collected by the authors and appears to be systematic. The extensive work by Tom Turner puts the South African data into the necessary global context. Collectively 100s of person years of work is referenced in total. Loss of biodiversity due to human influence will become increasingly important and milestone scientific evidence will be crucial for governments to act.

·

Basic reporting

The manuscript is well written and clear. A few grammatical changes are suggested.
I question if the terminology of “invasive” / “invasions” is appropriate in the title and text. It may be more appropriate to characterize H. perlevis as “introduced”, since the authors found “no evidence in South Africa that H. perlevis diplaced any species in low energy environments.
Line: 540: Invasive- ?perhaps, Introduced.
Line 617 refers to “introduced H. perlevis in California”

Background
What is the current state of knowledge on intertidal sponges in the study area?
Lines: 387 to 391 mention Tedania cf. scotia and T. cf. anhalens, are these native/ cryptogenic/introduced/ unknown?

Habitat and distribution
I am familiar with H. perlevis in Canada, a small isolated population, found on hard objects on a mudflat. Although, dna sequences are not available, the Canadian population (and others?) should be noted in Figure 1. (Harbo et al. 2021)
Lines 398-400, only in one location [in this study] …H. perlevis covered by sediment..
Lines: 538-539: Habitat and distribution: add Tolerant of sediment loads.

Vectors of introduction
Lines 581-583 relate to larval distribution. Mature, adult sponges may also have been transported on shells and other objects in aquaculture activities.
The introduction of H. perlevis in Canada was most likely associated with oyster aquaculture activities. This may have also been a vector in California, USA.

Experimental design

Meets objectives and criteria.

Validity of the findings

Discussion

Annual mean temperatures of native range and introduced distribution: suggests it is a cool/warm water species, limited by temperature. The short lived larvae in some locations, e.g. embayments, may be limited by other factors such as freshwater input, current patters that entrain larvae or?

………………..

Additional comments

Text comments

Line 86 (singular rather than plural) is the most distributed sponge…

Line 102: reorganize.
To date, genetic and morphological data support a broad distribution. Genetic studies have been conducted on the Atlantic coast….
Line 107 The species has also been described from morphological characters from New Zealand, …Canada (British Columbia)…

Thank you for the opportunity to read and to comment on this manuscript.

Rick Harbo, Research Associate
Invertebrate Zoology, Royal British Columbia Museum
675 Bellville Street, Victoria, B.C. Canada V8W 1A1
rmharbo1@gmail.com

Reviewer 3 ·

Basic reporting

Much of the paper fails to meet basic standards of a paper ready for submission to any journal. I provide detailed comments below.

Experimental design

Not applicable. Despite conclusions that would require experimental evidence, no experiments were done.

Validity of the findings

Please see review, below.

Additional comments

This 48-page manuscript genetically confirms that the sponge H. perlevis, long reported from South Africa, is correctly identified there (and is in Namibia as well). This paper could be reduced to a simple note for a journal such as Bioinvasion Records. A reader would presume from the title that the paper focuses on establishing (in this case, verifying) the record, but instead the paper attempts to also capture a global genetic and a global predictive distribution picture, all of which material could be bundled in a separate manuscript – that, or the title of the paper must be changed.

There are numerous challenges with this manuscript. As such, the paper was not ready for submission for peer review.

I’ll start with biogeography. Turner (2020) clearly established that this is a species widely distributed by human activity. Thus there’s no need at the outset to imply that the origin of its distribution in various parts of the world is uncertain (page 2, “but also occurs in” a number of locations, in juxtaposition to where it is thought to be introduced, where it is not thought? or was not thought? to be introduced), mediated by unknown vectors (line 114: “the vector is unknown” – although the vectors are very clearly suggested later, even starting on lines 139-140). Critically, the title of the paper suggests the species is native to Great Britain, but I can’t find clearly argued support for this in either this paper or in Turner 2020 – could it not be native to much of western Europe and even the northern Mediterranean? Are there temporal data that suggest it was introduced to southern Europe (Portugal and Spain) and the Mediterranean? – or could it have been introduced to England from these regions? Type localities are well known to not represent the native distribution of a great many species.

Confusion begins at line 102: Genetic and morphological data support a distribution in Portugal, Spain, and the Mediterranean, the Canary Islands, Florida and Argentina, California, and Japan, China, and Korea. The species has also been reported, the paper says, from New Zealand, Australia, West Africa, Namibia, South Africa, British Columbia, and the “Pacific coast of South America,” but no genetic data support these identifications. It is then said to have been introduced to the Mediterranean, Adriatic, and the Black Sea – but not to Portugal and Spain? A few lines later we learn it may be introduced (not is introduced, “may well be”) in California, Argentina, Korea, and China – but Japan is omitted. Could it possibly be native to California, Argentina, Korea, and China? (otherwise one would want to say, “its widespread presence today in …(these areas) … is explained by introductions from other areas” (not “may well be”).

Not mentioned is exactly what we know of the history of this species in Great Britain, flagged in the title of the paper. The implication in the introduction is, curiously, that the species is not confirmed from Great Britain, because it does not appear in the genetically confirmed list (only Portugal and Spain in Western Europe) – and yet it is presumed to be native to “southern England”, and we then find that it is regarded as confirmed from Ireland and Wales (lines 552-553) based apparently only on morphological data (but the reader would not know, in looking at this MS, that Turner 2020 has genetic data from Ireland).

(Background)

I first thought you might be using “South African” in an adjectival form, since the title of the paper omits Namibia, but that doesn’t seem to be the case. Why is Namibia omitted from the paper title?

Delete the last line of the “Background” summary. Adding H. perlevis “to the invasive list” is surely not the main concluding point of the paper (unless it’s reduced to a bioinvasion note), given the broader global context.


Main text:

Please delete the opening paragraph – this material begins a thousand papers on introduced species, and are not necessary.

Line 70 for “especially” read “including”. Invasions are especially common in estuaries, harbors, and ports, not on rocky intertidal shores.

Lines 74-78 Perna perna cannot, by definition, be a well-known example of a rocky shore introduction if it’s (line 78) cryptogenic in its purported non-native range.

Line 90 no need to repeat Montagu, 1814 – already in line 86

Line 90 delete “Northern Hemisphere” (there is no other England)

Lines 93-101 = are very much out of a place as the fourth paragraph of the Introduction; it almost appears as if this material was meant to be in the taxonomic section, or certainly somewhere else. A reader does not need to know at the very outset, for example, that the ectosomal skeleton is paratangential and the choanosomal skeleton is less dense.

Line 103 on the previous page the Adriatic Sea is treated as separate from the Mediterranean; here, it is a subset of the Mediterranean

Line 107 for “described” I’m guessing you mean largely “reported,” unless there is in fact a detailed morphological description of material from all of these places

Line 116 grammar: no commas in front of or after “H. perlevis”

Line 119 the “crumb of bread sponge” appears to be used for a number of different species, is that correct?

Line 120 “conspicuous in”, not “conspicuous to”

Line 121 for “early 1900” read “early 1900s” – but why not just say 1904?

Line 132 please add date of Awad

Line 138 read, “in low larval dispersal capacity” (clearly the species’ global distribution is not linked to its larval biology)

Line 153 The type material of caruncula can only be the original specimens (if they exist) of Bowerbank; Stephens ‘1915 specimens of caruncula are not “type material” by any definition, even sensu Stephens – they are simply the specimens he examined and gave a name to

Line 153 It’s important to add here, for clarity, that Stephens’ 1915 material was found at a museum and what it was preserved in, and thus was available for study and potential genetic workup – I’m also not finding this in the Methods

Line 164 do you mean this sponge sits directly in or on the sand, and not on a hard object (like shell fragments) in the sand?

Line 166 “while others are submerged during low”

Lines 174-175 what is the phrase “and was placed in bags for laboratory processing” doing here? – it seems to be a non sequitur following a habitat description?

Line 178 read, “Sponges were collected from intertidal rocky …”

Lines 179-180 how does “substratum” differ from “habitat … type” ?

Lines 181-183 does this mean that no range of variation is or will be reported for spicule dimensions?

Lines 213-214 “All the South African specimens had a high sequence similarity to H. perlevis” = these are Results

Line 215 “… from a single species with a global distribution …” – on the next page, this species is referred to clearly as a species complex

Lines 239-240 largely repeat what’s in lines 214-216

Line 254 using OBIS records would appear to conflict with a fundamental premise of the paper -- that records without genetic support cannot be verified – the very opening of the manuscript establishes this, stating that the presence of H. perlevis in South Africa is (was) in doubt “pending molecular confirmation.” How, then, could OBIS records be used? – if all of the OBIS records are based on genetic confirmation, then those records are all published elsewhere, and you don’t need the OBIS records. I also see that on lines 467-468 H. perlevis’ presence is accepted as valid in New Zealand, although listed earlier amongst locations that have not been verified genetically (lines 107-110). Again, there’s strong internal conflict here – at this point, is genetic data required or not to confirm H. perlevis in every case? Do the authors mean (lines 107-110 again) to suggest or imply that the recent report, for example, of its presence in British Columbia could be incorrect?

Lines 338-381 all of this material in this long section goes to Discussion – there are no “Results” here

Lines 376-378 grammar: “For example, kelp beds, rock lobster, cape sea urchins ….have established major distributional southeast expansions along the …”

Line 383 and following Notice that the fact that H. perlevis is not found north of Dwesa/Cwebe is repeated three times within 10 lines, suggesting that no careful editing of the paper was done prior to submission:

Lines 386-7: “Individuals of this species were not found north of Dwesa/Cwebe…”

Line 390: “However, H. perlevis, does not occur further north of Dwesa/Cwebe.”

Lines 396-7: “… it is absent from the subtropical province of the Agulhas system north of Dwesa/Cwebe”

And even a fourth time!:

Line 599: “but it is absent north of Dwesa/Cwebe and the subtropical province of the Agulhas system”

Lines 398-399 Re: “In only one location (Strand) we found …” (in sand) … following this is a sentence, introduced with “Apart from this, …” which is relative to where the sponge is found in areas with high nutrient concentrations. This is a non sequitur; the “Apart from this” ( = the sand habitat) has nothing to do with whether it is or is not in areas of high nutrient concentration.

Line 402 Stephens, not Stephenson

Line 404 reported or described?

Line 423 “etc.” has no meaning here – either say something like, (for example, Portugal and California), or add more locations, or refer to a figure or table

Line 466 “Hymeniacidon perlevis is unusual in that it has a very wide geographic distribution ….” -- please clarify: unusual for a sponge? Or unusual for invertebrates in general? Unusual for an introduced species?

Lines 447 and 456 … H. perlevis is in the Caribbean
(but)
Line 484 … the modeling produced no suitable habitat for the Caribbean, which is describe as “surprising”

= I’m not finding (and apologies if I missed this) an explanation as to why the model did not predict the Caribbean, since the species is in fact found in the Caribbean. Does this suggest that some of the model parameters were insufficient?

In general, one must ask if the habitat modeling that was undertaken revealed anything significant that simple inspection of its geographic range (based on temperature regimes) would not? – on lines 591-593 we learn the “importance of sea surface temperature in shaping the distributional range of H. perlevis” – surely one can determine this by simple inspection of isotherms, without modeling? It seems clear from the distribution of the species that it is a cool / warm temperate species (line 619) – please explain why modelling was necessary?

Lines 537-538 Scattered throughout the paper are casual comments about the habitat and ecology of this species. These should be brought together in one place. Here, for example, is the comment, “Often exposed to direct sunlight.” [is this comment relative to southern Africa specifically?] (Earlier [lines 399-400] we learned that this species was also found in places … with high nutrient concentration” -- one of the casual habitat comments)

Lines 544-547 “Comparisons of the morphological features of … H. perlevis … with Hymeniacidon caruncula Bowerbank 1858 sensu Stevens (1915) showed that it is similar to H. caruncula …” -- But it was already established that caruncula is perlevis, so did this mean to say, “ … similar to H. perlevis” ??

Lines 552-553 Here it’s noted that the spicule size range of SA perlevis “overlaps with specimens from Wales, Korea and Ireland” … while this is of interest, we are told later that this type of evidence does not establish conspecificity in the absence of genetic data; here, the material from Wales has not been genetically assessed?

Line 556 Again, “etc.” has no meaning here.

Line 560 there is a quotation mark between <is> and <A thick,> --- where is the end of this quotation?

Line 580 “human-mediated gene flow is likely to play an important role” = given that it is established that the larvae can’t disperse for any long distances, do we need the modifier “likely” here? -- if larval ecology and biology cannot explain the widespread distribution of this species, and human-mediated assistance is (only) “likely”, what are the other alternatives?

Line 581 should this read, “Because larvae have a very limited chance of long-distance survival …” – or do you mean survival overall?

Lines 582-583 same question as above – do we need “the most likely vectors” here? If movement of this sponge by ocean-going vessels is an insufficient mechanism to fully explain the distribution (necessitating “most likely”), what are other mechanisms?

Lines 595-599 = repeats the material on lines 392-397 … in fact, this appears to be the exact same sentence! – what happened here?

Lines 604-605 There appears to be no other reference in this paper to this species’ “expansion on the south coast over the last 80 years.” What is the evidence for this species expansion over this period of time? What are the baseline date? 1904 is the first record – more than 100 years ago. Stephenson’s surveys began in 1932 – about 90 years ago. Do the cited references (Stephenson 1938, 1944, and 1948, and Day 1974) clearly outline evidence for this expansion from the first detected populations? [Note that Stephenson appears as 1939 in the references] Is there evidence of absence of H. perlevis in some regions such that it wasn’t simply overlooked earlier, and thus only appears to have expanded?

Line 608 should this read, “This limit could be explained by the fact that the area ..”

Lines 610-611 if the larvae have a very short life in the water column, then ocean currents could not play a role in the species’ distribution, in which case how could there be an “oceanographic barrier to larval dispersal” ?

Line 616 there is no Anonymous in the References

Line 632 by “Indo Pacific” shipping do you mean shipping through the Indo-Pacific (as a route), since the species does not occur in the Indo-Pacific?

Lines 639-640: “the observed pattern cannot be maintained by natural dispersal and gene flow at such a broad scale”
Please note Line 576 ! = “the observed pattern cannot be maintained by natural dispersal and gene flow at such a broad scale” – lines 639-40 exactly repeat this

Line 656: “along the Pacific Ocean” – what does this mean? (“in the Pacific Ocean”?)

Lines 659-662 “The successful invasions of H. perlevis in South Africa, Argentina, North America, Europe, and Japan does not suggest that this species may pose a threat to intertidal communities in cool/warm temperate areas.”

As the species is native to at least part of Europe, this should be clarified.

Grammar: “invasions do …” (plural)

But, most importantly, the first part of the sentence does not logically permit the conclusion in the second part of the sentence, unless the word “not” was inserted by mistake. The fact that this sponge has invaded all of these regions does not suggest anything about its impacts or the threats it posed or poses to intertidal communities. These concepts are unrelated. A great many species have successfully invaded all of these regions, and have had profound impacts on the resident communities in marine, freshwater, and terrestrial habitats. In fact, it’s hard to understand what this sentence could mean. This sentence should be deleted – it makes no sense.

Lines 661-662: “There is no evidence in South Africa that H. perlevis displaced any species in low energy environments, like M. galloprovincialis.”

And, in the absence of such evidence, we can also conclude that there is no evidence that H. perlevis has NOT displaced any species. What is the negative evidence? Historical studies showing that the before invasion - after invasion diversity (richness) did not change? Historical studies demonstrating that the relative abundance of species before-after H. perlevis’ arrival did not change? Experimental studies? This statement should be deleted unless quantitative or experimental evidence can be cited that H. perlevis did NOT displace any species.

Lines 662-664: “The findings of this study emphasise [sic] that similar environmental conditions appear to facilitate successful marine introductions.”

= Please delete this sentence. This has been a standard ecological observation relative to introduced species for well over 100 years, and appears in virtually every ecology textbook. It does not need to be emphasized.

Lines 671-672 “ … where it provides a habitat for numerous other species that colonize it”
= This statement comes “out of the blue,” with no precedent in the Methods, Results, or Discussion, similar to the casual comments that H. perlevis is often exposed to direct sunlight or is found in areas of high nutrient concentration – both sans data. Why does this statement about this sponge providing habitat for other species appear first and only here in the Conclusions? Bundle all of these (anecdotal?) ecological-biological observations together in the appropriate place in the paper.

Lines 672-673 “There is no indication, however, that the species is displacing native species.”
= As above, this would require some evidence, for which none has been offered.

Line 675 Given the conclusions / suggestions that this sponge does not pose a threat and
has not displaced native species (both of which I strongly urge be deleted), why does the last thought in the paper end with the denouement that surveys “would provide a better indication of the invasion’s severity” ? (which we’ve just learned probably isn’t really a big deal).

Acknowledgements

Note that the term “Senior Author” in some countries means the last author.

Figures:

Figure 2 (and caption)
Do you mean black dots, not white dots?

De-coding the locations should not be relegated to supplementary material! Please provide the full names of each 2-letter site code in the caption. Dwesa/Cwebe should be clearly marked on the map, given its repeated mention and importance in the paper – one might have thought there might be at least a D/C on this map?
The figure caption says “Surveyed locations are described in Supplementary Table 1” – I can’t find any of the 2-or 3-letter codes shown on the Figure 2 map de-coded in Table S1 –are they somewhere else?

---

## Round 0.2 · Major Revisions

Thank you for the substantial revisions made to your manuscript. Reviewer 3 has provided another thoughtful and thorough report on the revised manuscript. Please address their comments in a further revision. I'd suggest dropping the "There and back again:" from the title.

Reviewer 3 ·

Basic reporting

please see below

Experimental design

please see below

Validity of the findings

please see below

Additional comments

Review: peerj-72071: “There and back again ….”

The manuscript has been substantially revised, but key aspects still require attention.

Re: Title of paper:

Sorry, two things:

(1) “There and back again”: I appreciate the allusion to Tolkien, The Hobbit, and the adventures of Bilbo Baggins: it’s true that this sponge has wandered around the world, but, that said, the conclusions of the manuscript do not support either a “back” or “back again” – back to where, for example? The manuscript clearly establishes that a dispersal history cannot yet be elucidated based upon either biogeography or the haplotype picture. Using the phrase “There and back again” only serves to confuse, not clarify, the picture.

(2) You introduce the word / concept “cryptogenic” in the title – but I can’t seem to find that that word is used anywhere else in the paper.

Global Distribution:

Challenging is that this manuscript still appears to fail to transparently make it clear whether molecular work is or is not needed to identify this sponge. On the one hand, we read that the identity of this sponge in South Africa was “in doubt pending molecular confirmation” and despite the previous South Africa records, “genetic confirmation of the species identity is needed within southern Africa.” Fair enough it seems. Supporting this are statements such as “Hymeniacidon species … exhibit a high degree of morphological similarity, making it difficult to recognize them solely by morphological characters. Traditional morphological characters … are insufficient to distinguish species in this complex.”

In contrast, it appears that many records from around the world have been fully accepted by the authors as this species based only on morphological evidence: “ … the species has been found in temperate waters of all major ocean basins …” … “ (it) has a very wide geographic distribution ranging from the Northern Hemisphere to Argentina, as well as South Africa to the middle latitudes of New Zealand.” These global occurrences are then plotted in Figure 1, which is labelled, “Global occurrence records of Hymeniacidon perlevis.”

But clearly when we examine the paper and the supplementary files, many of these global records are NOT verified genetically, leaving us with a strong internal conflict – to verify South Africa, molecular work was required. To accept many other global records, molecular work was not required. The key sentence to accept all these non-molecular records appears to be this: “The location data were checked and verified by Toufiek Samaai and only valid and substantiated occurrence records were included in this study. Bias and unverified data were excluded.”

Again, fair enough to identify a filter – but what the filter is / was is unstated! In the absence of genetic data, how were all the global records that were based only on morphology “verified”, if the species cannot be reliably identified based on morphology??

A conclusion one might draw is that the paper unnecessarily struggles with defending the need to do molecular work for South Africa – a defense not required, as evidenced by the authors’ clear willingness to accept many non-genetically-based records as valid H. perlevis records. The simplest solution (again, apologies if I’m misreading things here!) is to drop the sentences that state that molecular work was a sine qua non to accept the South African record, and simply say that this paper now contributes, having had the opportunity to analyze SA populations, to furthering our knowledge of the genetic picture of this now-global species. The MS then needs to attend to the internal conflict that Hymeniacidon species cannot be reliably identified morphologically.

I would urge that Figure 1 be re-done and coded in two distinct ways: populations that have been genetically characterized, and those populations (locations) where the identification is based (apparently) on morphological grounds only.

Additional comments:

Abstract:
The abstract should include an important conclusion – that the species might be native to Europe or southern Africa and that while apparently clearly introduced in many places around the world, it is thus (frustratingly) cryptogenic in both Europe and southern Africa.

Line 38 “rocky” not “Rocky”

Line 39 what does the phrase “discuss possible underlying mechanisms” mean? Mechanisms controlling the species’ global distribution? or ?

Lines 68-69 “and it has 29 synonymized names ….. (de Voogd et al., 2022).”

Gosh no! -- there are only about 18 synonymized scientific names listed in de Voogd et al. 2022 (and repeated in WoRMS). “29” is a misreading of the nomenclatural intricacies. Study the list carefully! Notice that one of your 29 synonyms is H. perleve, a rejected misspelling, and another one of the “synonyms” is Sponge perlevis (the original form of the name)! The names caruncula, sanguinea, uniformis, simplissima, aurea, mammeata, and others are all repeated once or twice in this list, due to different nomenclatural renditions and constructions. The WPD (and WoRMS) bundle these as “synonymized names” for simplification, when in fact many of them are simply part of the species’ chresonymy.

Line 69 “…. from approximately 100 different locations throughout the world (de Voogd et al., 2022).”

This needs to be clarified: The global map presented (Figure 1) does not appear to show anywhere near 100 different locations, although it’s hard to tell given the occasional density or confluences of the locations plotted.

Line 73 p. for pp.?

Line 80 “is now” not “it is now”

Line 122 “and associated management implications” – no longer really delved into in this revision, correct?

Line 130 “The holotype of”

Line 131 “is persevered in” (not, “are preserved in in”)

Line 400 “Two fragments” [plural]

Lines 413-417 and lines 445-448 – let’s combine these two now-separated statements about colour

Line 415 Please clarify what “During sampling the sponge’s colour may vary; it may become dark patchy brown or seem greenish” – how does the sponge’s colour change while sampling?? – do you mean after it’s collected and put in preservative, or ?

Line 429 “Status. Introduced.” -- this appears to be a lapsus, since the paper now concludes that the species may be native to South Africa !

Lines 460-461 “Ngwakum … previously deemed this to be improbable” – what does “this” refer to? that it is the most common sponge? Or that it was H. perlevis?

Line 460 Ngwakum et al. 2020 – or 2021?

Line 523 “Due to” (not “Due of”)

Line 545 “Europe and/or South Africa” – are you suggesting that H. perlevis could be native to BOTH Europe and South Africa? … or do you mean “or” ? If it is native to both regions, how did that happen biogeographically and over evolutionary time?

---

## Round 0.3 · accepted · Accept

Thank you for addressing Reviewer 3's comments. I am now pleased to accept your manuscript for publication.

Reviewer 3 ·

Basic reporting

see below

Experimental design

see below

Validity of the findings

see below

Additional comments

The authors have addressed all of my concerns.